# NAMPT Over-Expression Recapitulates the BRAF Inhibitor Resistant Phenotype Plasticity in Melanoma

**DOI:** 10.3390/cancers12123855

**Published:** 2020-12-20

**Authors:** Valentina Audrito, Vincenzo Gianluca Messana, Enrico Moiso, Nicoletta Vitale, Francesca Arruga, Lorenzo Brandimarte, Federica Gaudino, Elisa Pellegrino, Tiziana Vaisitti, Chiara Riganti, Roberto Piva, Silvia Deaglio

**Affiliations:** 1Cancer Immunogenetics Lab, Department of Medical Sciences, University of Turin, 10126 Turin, Italy; vincenzo.messana@unito.it (V.G.M.); nicoletta.vitale@unito.it (N.V.); francesca.arruga@unito.it (F.A.); lorenzo.brandimar@edu.unito.it (L.B.); federica.gaudino@unito.it (F.G.); tiziana.vaisitti@unito.it (T.V.); 2Koch Institute for Integrative Cancer Research, Massachusetts Institute of Technology, Cambridge, MA 02139, USA; emoiso@mit.edu; 3Broad Institute of Harvard and MIT, Cambridge, MA 02142, USA; 4Department of Molecular Biotechnologies and Health Sciences, University of Turin, 10126 Turin, Italy; elisa.pellegrino@unito.it (E.P.); roberto.piva@unito.it (R.P.); 5Department of Oncology, University of Turin, 10126 Turin, Italy; chiara.riganti@unito.it

**Keywords:** metastatic melanoma, BRAF inhibitors resistance, BRAF, metabolic reprogramming, NAD, NAMPT, MAPK, oncogene, mesenchymal phenotype, stemness

## Abstract

**Simple Summary:**

Malignant melanoma (MM) is the most fatal skin cancer due to its high metastatic potential. Treatment strategies are dramatically changing due to the introduction of BRAF/MEK inhibitors (i) and immunotherapy; however, multiple resistant mechanisms rapidly occur including metabolic rewiring. This study aimed to establish the driver role of the nicotinamide adenine dinucleotide (NAD)-biosynthetic enzyme nicotinamide phosphoribosyltransferase (NAMPT) in BRAFi resistance development. We defined that NAMPT over-expressing MM cells were strikingly similar to cells that acquired resistance to BRAFi in terms of growth, invasion, and phenotype plasticity. These findings confirmed NAMPT as a key factor in melanoma progression and in the onset of BRAFi resistance in melanoma patients, opening new therapeutic possibilities for this subset of patients.

**Abstract:**

Serine–threonine protein kinase B-RAF **(***BRAF)*-mutated metastatic melanoma (MM) is a highly aggressive type of skin cancer. Treatment of MM patients using BRAF/MEK inhibitors (BRAFi/MEKi) eventually leads to drug resistance, limiting any clinical benefit. Herein, we demonstrated that the nicotinamide adenine dinucleotide (NAD)-biosynthetic enzyme nicotinamide phosphoribosyltransferase (NAMPT) is a driving factor in BRAFi resistance development. Using stable and inducible NAMPT over-expression systems, we showed that forced NAMPT expression in MM *BRAF*-mutated cell lines led to increased energy production, MAPK activation, colony-formation capacity, and enhance tumorigenicity in vivo. Moreover, NAMPT over-expressing cells switched toward an invasive/mesenchymal phenotype, up-regulating expression of ZEB1 and TWIST, two transcription factors driving the epithelial to mesenchymal transition (EMT) process. Consistently, within the NAMPT-overexpressing cell line variants, we observed an increased percentage of a rare, drug-effluxing stem cell-like side population (SP) of cells, paralleled by up-regulation of ABCC1/MRP1 expression and CD133-positive cells. The direct correlation between NAMPT expression and gene set enrichments involving metastasis, invasiveness and mesenchymal/stemness properties were verified also in melanoma patients by analyzing The Cancer Genome Atlas (TCGA) datasets. On the other hand, CRISPR/Cas9 full knock-out *NAMPT* BRAFi-resistant MM cells are not viable, while inducible partial silencing drastically reduces tumor growth and aggressiveness. Overall, this work revealed that NAMPT over-expression is both necessary and sufficient to recapitulate the BRAFi-resistant phenotype plasticity.

## 1. Introduction

*BRAF* V600E leads to constitutive activation of BRAF and, consequently, of the RAF-MEK-ERK signaling cascade, promoting cell proliferation and survival while inhibiting apoptosis, ultimately driving cancer growth [1,2]. As a result, several selective BRAF inhibitors (BRAFi) are now being clinically used in combination with MEK inhibitors with significant results, particularly for metastatic melanoma (MM) patients [3,4,5]. The clinical benefit of this treatment is limited by the onset of resistance, underlining the need for additional therapeutic targets. Resistance is accompanied by paradoxical activation of the MEK/ERK pathway, bypassing the signaling block caused by the inhibitor, with mutations developing downstream of the signaling block. Resistance is often driven by reprogramming the cell metabolism, epigenetics and gene expression, all of which lead to a dynamic deregulation of differentiation, mesenchymal, and stemness transcriptional programs [6,7,8]. Overactivation of the BRAF oncogenic pathway drives these reprogramming processes [8,9,10,11].

The metabolism of nicotinamide adenine dinucleotide (NAD), an essential redox cofactor needed for mitochondrial respiration and a signaling molecule, is altered during cancer progression [12]. *BRAF*-mutated MM contains higher levels of NAD, which drastically decrease in response to BRAFi/MEKi treatment, increasing in BRAFi-resistant cells [13,14]. Even though NAD can be generated through multiple pathways, increased production of NAD in MM relies exclusively on nicotinamide phosphoribosyltransferase (NAMPT) activity [15,16], the rate-limiting enzyme in the synthesis of NAD from nicotinamide [13,17].

In this study, we definitively demonstrated that the over-expression of NAMPT is not a consequence of BRAFi resistance but a key event both necessary and sufficient to recapitulating the BRAFi-resistant phenotype. In fact, BRAF-mutated NAMPT over-expressing MM cells are strikingly similar to cells that have acquired resistance to BRAFi in terms of growth, invasion and phenotype plasticity, including mesenchymal and stem properties. Overall, our results support the hypothesis of a driving role of NAMPT in melanoma progression and the clinical use of NAMPT inhibitors in combination with BRAFi/MEKi to avoid onset resistance.

## 2. Results

### 2.1. NAMPT Over-Expression Enhances Metabolic Fluxes

Based on our previous data, which indicated that BRAFi-resistant cells over-express NAMPT, we wondered what the effects were of NAMPT over-expression in BRAF V600E mutated melanoma cells. To answer this question, we used BRAF V600E melanoma cell lines (A375 and M14) modified to stably over-express NAMPT (NAMPT/GFP) and functionally studied their behavior by comparing them to control GFP and BRAFi-resistant (/BiR) cells.

As expected, based on increased NAD levels [13], NAMPT over-expressing cells increased both aerobic glycolysis and oxidative phosphorylation (OXPHOS), demonstrating an energetic phenotype (Figure 1A,B). Consistently, the glucose transporter SLC2A1/GLUT1 was significantly up-regulated in NAMPT/GFP cells compared to GFP cells, in both cell lines, as shown in mRNA and protein levels (Appendix A). Likewise, both glyceraldehyde 3-phosphate dehydrogenase (GAPDH) and lactate dehydrogenase (LDH), as well as the activity of electron transport chain (ETC), evaluated as the electron flux from complex I to complex III, were increased in NAMPT/GFP cells compared to the control. However, two lines maintained their original differences with a more evident Warburg phenotype in M14 NAMPT/GFP cells, where the activity of GAPDH and LDH is higher, and a marked increase of ETC flux in A375 NAMPT/GFP (Figure 1C,D), as occurred in/BiR cell variants [13].

M14 and A375 NAMPT/GFP also displayed a relevant increase of the pentose phosphate pathway (PPP) rate and of the activity of the glucose-6-phosphate dehydrogenase (G6PD, rate-limiting enzyme in this pathway, Figure 1E), which was in line with recent observations made in BRAFi-resistant MM cells [18].

### 2.2. NAMPT Over-Expression Boosts Tumor Growth Capacity In Vitro and In Vivo and ERK Phosphorylation

In keeping with highlighted metabolic performances, NAMPT/GFP cells were able to generate more colonies compared GFP clones over a 12-day period. Importantly, culture in the presence of BRAFi almost completely inhibited colony formation in A375 and M14 cells, while NAMPT/GFP cells were intrinsically resistant to the BRAFi Dabrafenib (Figure 2A,B). Results were confirmed by using Vemurafenib (Appendix A) and stopping the experiment after 1 week. In line with this hypothesis, NAMPT/GFP cells displayed more intense phosphorylation of ERK1/2 in response to serum stimulation after starvation (Appendix A), similarly to what we observed in/BiR cells [13,19]. Soft-agar colony formation confirmed increased growth capacity of NAMPT/GFP cells, with markedly bigger clones compared to control cells (Figure 2C,D).

To measure the in vivo growth of NAMPT over-expressing cells, we turned to an inducible model where NAMPT over-expression was triggered by exposure to doxycycline (DOX). These cells were injected in NOD/SCID/IL2Rγ-/-(NSG) mice, and animals were treated or not with DOX, added to the drinking water (Appendix A), to obtain differential NAMPT expression (Figure 2E,G). NAMPT over-expression in vivo increased tumor formation capability in both M14 and A375 cell lines (light blue line + DOX vs. black line-DOX), as highlighted following tumor growth kinetics over 4–5 weeks and analyzing tumor masses (Figure 2F,H).

### 2.3. NAMPT Over-Expression Promotes Invasion and Acquisition of Mesenchymal Features

Both M14 and A375 NAMPT/GFP variants showed a more aggressive behavior, when examined using classical invasion assays performed in Matrigel, with a marked increase of the invasion index (Figure 3A), in line with higher plasticity. Accordingly, the activity of matrix metalloproteinases (MMP), enzymes implicated in invasion and metastasis in MM [20], was significantly higher in NAMPT/GFP variants of both cell lines compared to GFP. Gelatin zymography assays detected increased total gelatinolytic activity in the NAMPT/GFP cells at molecular weights compatible with MMP-9 (92 kDa), MMP-2 (72 kDa), MMP-1 and MMP-3 (55–45 kDa, Figure 3B). The third data substantiating the acquisition of a more aggressive behavior by NAMPT/GFP cell lines was the over-expression of integrin alpha 3 (ITGα3/CD49c), a molecule previously associated with invasive features and a drug-resistant signature [21,22] (Figure 3C and Appendix A).

It is reported that MM cells that acquire BRAFi resistance lose their differentiation, increase their metastatic potential and undergo a transition from an epithelial to a mesenchymal (EMT) phenotype [23,24]. This transition is regulated by the activation of transcription factors, such as TWIST and ZEB1, which drive expression of vimentin/VIM and CD56/NCAM, well-known EMT markers [23,25]. Consistently, NAMPT/GFP cells showed higher levels of TWIST and ZEB1 and of target proteins VIM and NCAM (Figure 3D,E and Appendix A). These effects were more evident in M14 vs. A375 variants, suggesting that in cells that are intrinsically less invasive, the effects of NAMPT over-expression are more marked.

We then asked whether the observed effects of phenotype plasticity could directly be caused by the NAMPT-controlled enzymatic reaction, leading to increased NAD availability. To this end, we used the product of the NAMPT reaction (i.e., nicotinamide mononucleotide (NMN)), which was added to the cell culture media, as it is water soluble. For 24 h, the culture of the GFP cells in the presence of 1 mM NMN was followed by significant up-regulation of ZEB1 in both M14 and A375, while up-regulation of its target VIM was evident mostly in M14 cells (Appendix A). Consistently, cells cultured in the presence of NMN (1 mM for 24 h) migrated better than cells cultured without NMN. The effect was especially evident for A375, which migrated constitutively better than M14 (Appendix A). However, in absolute terms, the effects obtained when using NMN in GFP cells were generally lower than the effects scored by NAMPT/GFP cells.

### 2.4. NAMPT Over-Expression Forces Stemness Properties

Starting from the observations that (i) the EMT process is associated with the generation and maintenance of cancer stem cells (CSC) [26] and that (ii) the BRAFi-resistant phenotype is accompanied by increased stemness properties [27], we wondered if NAMPT over-expression induced a stem-like phenotype.

As a first observation, NAMPT/GFP cells showed markedly increased expression of the multidrug resistance protein 1 (MRP1) encoded by the *ABCC1* gene, originally discovered as the leading mechanism of multidrug resistance in tumor cells [28] (Figure 4A,B). Confocal microscopy indicated that MRP1 expression was restricted to a subset of cells, as expected (Figure 4B). In fact, the ATP-binding cassette (ABC) transporters, such as MRP1, were expressed by a wide variety of stem cells and were markers of the side-population (SP) phenotype. SP cells were defined as the cell fraction that efficiently excluded the fluorescent dye and anticancer drugs, representing true chemo-resistant cells [29]. In line with MPR1 over-expression in NAMPT/GFP cells, we observed a significant expansion of the percentage (%) of SP, measured as the percentage of negative cells after dye cycle violet (DCV) staining (Figure 4C). In addition, the percentage of cells positive for the stem cell marker CD133, widely used to characterize and isolate the putative melanoma CSC both in vitro and in vivo [30], increased significantly in NAMPT/GFP compared to control GFP cells (Figure 4D).

Consistently, when stratifying gene expression data from the Cancer Genome Atlas (TCGA), based on *NAMPT* expression, it was apparent that tumors with higher *NAMPT* expression were more associated with gene signatures of RAS/MAPK signaling, metastasis, invasiveness, EMT and stemness compared to tumors with lower *NAMPT* expression (Figure 5A). Analyzing specific genes involved in phenotypic plasticity, the TCGA dataset confirmed the direct and significant correlation between expression levels of molecules connected to EMT (*ZEB1*) and stemness (*YAP1, ABCC4*) molecules, as well as *NAMPT* in melanoma patients (Figure 5B). Notably, we dissected the cohort of melanoma patients in primary (blue dots) and metastatic (red dots). This analysis revealed that the correlation is more significant in primary samples (Figure 5B).

### 2.5. NAMPT Silencing Reverts Aggressiveness of BRAFi-Resistant Cells

To confirm the role of NAMPT as a driver of MM progression and drug resistance, we silenced NAMPT in/BiR cells using two different strategies: (i) CRISPR/Cas9 to obtain a full knock-out (KO) and ii) inducible shRNAs. To generate full NAMPT/KO, two different guides (sgRNA#2 and sgRNA#7, Appendix A), both targeting exon 1, were designed and cloned into the pX458 vector expressing the Cas9 and a GFP tag. shRNA-Cas9-GFP plasmids were then used to transfect A375/BiR cells, together with the Cas9-GFP empty vector (Cas9/empty-V) used as the control. GFP^+^ cells were sorted ~30 h after transfection and cultured for further functional experiments. The efficiency of the Cas9 enzyme was determined using the Surveyor assay for the A375/BiR Cas9/sgRNA#7 and A375/BiR Cas9/empty-V as control (Appendix A). NAMPT silencing was confirmed by the Western blot analysis (Appendix A). Approximately 24 h after sorting, NAMPT/KO cells displayed a sudden change in morphology with loss of epithelial-like shape compared to control cells. Moreover, 48 h after sorting cells started to die and were completely detached from plastic after 72 h, as documented by light microscopy analysis (Appendix A). To determine whether cell death was due to NAD deprivation, 0.5 mM of NMN was supplemented to the culture of sgRNA#7 cells immediately after sorting and after 16 h, with a marked rescue of cell viability and epithelial morphology (Appendix A). While these data are in line with the notion of a vital role of NAMPT in MM resistant cells, and in general in mammalian cells [31], they made it impossible to generate stable NAMPT/KO cells for functional characterization. For these reasons, we decided to use a second approach, building a cellular model where NAMPT was silenced via an inducible system. We expressed two different validated shRNAs targeting NAMPT (shA NAMPT, shC NAMPT, Appendix A) [32] and a shRNA control sequence (shCTRL) in both M14/BiR and A375/BiR cell lines under the control of the DOX-regulated transcriptional repressor tTR-KRAB (TTA). Exposure of the cells to DOX (1 µg/mL for 24–48 h) led to an ~80% reduction of NAMPT in both cell lines, as highlighted by RT-PCR and Western blot analysis (Appendix A). No compensatory activation of other NAD-biosynthetic enzymes (NBEs: NAPRT, NMRK1 and QPRT), which remained unmodulated and expressed at very low levels, could be determined by RT-PCR (Appendix A).

NAMPT silencing significantly decreased the colony-formation capability over a period of 10–12 days (Figure 6B,C) of/BiR cells in both cell lines and with both shRNA sequences. Mice subcutaneously injected with M14/BiR TTA shA NAMPT and A375/BiR TTA shA NAMPT cells, and treated with DOX (0.1 mg/mL biweekly), starting 24 h after the injection (Appendix A), showed marked reduction in tumor growth (tumor volume and weight, Figure 6D). Control mice injected with shCTRL cells and similarly treated with DOX did not display any differences compared to/BiR cells. Additionally, NAMPT knock-down led to a significant inhibition of the invasive properties of/BiR cells in both cell lines and with both shRNA sequences (Figure 7A,B). Lastly, we found a decrease of mesenchymal phenotype typical of aggressive and invasive MM [23,27]. Expression levels of two key transcription factors (i.e.*,* TWIST and ZEB1) and their target vimentin/VIM were reduced in NAMPT silenced cells (upon 24 h of DOX exposure) compared with shCTRL, as showed by RT-PCR in both/BiR TTA cell line variants (Figure 7C).

Taken together, these results clearly show that NAMPT is a driving protein in mediating aggressive and drug-resistant MM phenoype.

## 3. Discussion

*NAMPT* gene expression is finely-tuned at the transcriptional level by the BRAF oncogenic pathway [13,14]. Consistently, in BRAFi-resistant melanomas, NAD metabolism is increased through the selective up-regulation of the nicotinamide pathway via NAMPT activity [13,17].

To understand the functional role of NAMPT in the BRAFi resistance program and melanoma aggressiveness, we generated MM cell lines that over-expressed NAMPT in a stable or inducible way. Importantly these MM cell lines carry the *BRAF* V600E oncogene, but are not resistant to its inhibition. Over-expression of NAMPT was followed by (i) activation of metabolic pathways, including glycolysis, OXPHOS and also PPP, recently associated with drug resistance [18]; (ii) acquisition of intrinsic resistance to BRAFi; (iii) increased colony-formation capacity, also in 3D, and tumor growth in vivo; (iv) increased phosphorylation of ERK1/2, further sustaining oncogenic signaling; (v) increased invasive features including up-regulation of adhesion molecules and MMPs activity; (vi) activation of an EMT program, up-regulating key transcription factors, such as ZEB1 and TWIST1; (vii) increased expression of stem cell markers and of cells percentage with CSC-like features. Importantly, all these features define the BRAFi-resistant phenotype indicating that NAMPT over-expression per se recapitulates a resistance signature. A connection between NAMPT activity and invasive/stemness properties was recently reported in leukemia [33], glioma [34], colon [35] and breast [36] cancers. Our data supports the hypothesis that the phenotype plasticity of MM (oncogenic/invasive state ZEB1^high^/TWIST1^high^) observed in *BRAF* mutated patients [23], responsible for disease progression and drug resistance [6,37], is recapitulated, at least in part, by NAMPT over-expression. An open question concerns the mechanisms through which NAMPT exerts its “oncogenic” effects. One possibility is that the enzymatic activity is critical, by leading to a (documented) increase in net intracellular levels of NAD, thereby affecting activity of NAD-dependent enzymes, including sirtuins, both in the cytosol and/or directly in the nucleus [38]. In fact, addition of NMN, the enzymatic product of the NAMPT reaction, to M14 or A375 cells increased expression of mesenchymal markers and induced a more invasive phenotype. While these effects were measurable and significant, they remained anyway lower than what scored by M14 NAMPT/GFP and A375 NAMT/GFP cells, indicating either that exogenous administration of the product is less efficient or that there are additional enzyme-independent mechanisms of action of NAMPT. In this context, we documented presence of eNAMPT in the supernatants of both cell lines, with increasing concentrations in BiR cells [39] and also in the supernatants of NAMPT-overexpressing cells (not shown). Extracellular NAMPT signaling could take part in determining the aggressive phenotype, particularly considering that its putative receptor TLR4 [40] is highly expressed in these cells.

Overall, these results support the idea to consider NAMPT a novel actionable therapeutic target in MM. In the past, NAMPT inhibitors (NAMPTi) showed modest anti-tumor activity mostly because NAMPT block was bypassed by compensation through other NAD-biosynthetic pathways, such as that regulated through NAPRT and NRK [41,42,43]. Our previous data showed that treatment of melanomas with NAMPT inhibitors (NAMPTi) led to a metabolic crash decreasing NAD and ATP levels and ultimately inducing apoptosis in vitro and tumor regression in a xenograft model [13]. Accordingly, in this study we show that *NAMPT* knock-out in/BiR cells induces rapid cell death, while incomplete silencing leads to a marked inhibition of proliferation and loss of aggressive/invasive features. Altogether, these results are in line with a fundamental role of NAMPT/NAD in cellular metabolism and viability of mammalian cells, as also inferred by its widespread expression in all mammalian tissues [44] and by the notion that *NAMPT* gene deletion in mice is embryonically lethal [31]. From a translational perspective, we demonstrate that there is no compensatory upregulation of the other NBEs (NAPRT, QPRT and NRK) in response to NAMPT repression, supporting the possibility of clinical use of the inhibitors, in combination with BRAFi/MEKi, for this subset of cancer patients, addicted to NAMPT activity. Further studies are needed to confirm the therapeutic potential of this drugs combination.

Overall, the results of this work revealed that NAMPT over-expression is a key event in mediating drug resistance and in increasing aggressive features of melanomas opening to the hypothesis of a direct oncogenic role for this enzyme in melanoma tumorigenesis.

## 4. Materials and Methods

### 4.1. Reagents and Antibodies

The full list of reagents and antibodies used is in Appendix A.

### 4.2. Cell Culture

MM cell lines used were M14 and A375 BRAF V600E-mutated, sensitive or resistant to BRAFi, previously available in the lab [13]. Cell lines were tested to confirm lack of mycoplasma contamination. These cell lines were cultured as detailed in Appendix A.

### 4.3. Stable NAMPT Over-Expression

Stable NAMPT over-expression was obtained as described [13]. Doxycycline-dependent inducible NAMPT over-expression was obtained by lentivirus infection as detailed in Appendix A.

### 4.4. Inducible NAMPT Silencing (shRNA NAMPT)

Two different shRNAs glycerol stocks were purchased from Sigma, Milan, Italy (TRCN0000116177 = shA, TRCN0000116180 = shC), and previously validated [32], together with one control shRNA (SHC002 = shCTRL). Inducible shRNA NAMPT was achieved exploiting the tTRKRAB system, as detailed in Appendix A. Silenced cells were referred as A375/BiR TTA and M14/BiR TTA.

### 4.5. CRISPR-CAS9 NAMPT Knock-Out (KO) Cells

CRISPR/Cas9 *NAMPT*/KO was prepared as previously published [45] and detailed in Appendix A.

### 4.6. RNA Extraction and Quantitative Real-Time PCR (qRT-PCR)

RNA was extracted using RNeasy Plus Mini kit or QIAzol protocol (Qiagen, Milan, Italy) and converted to cDNA using the High Capacity cDNA Reverse Transcription kit (Thermo Fisher Scientific, Monza, Italy). qRT-PCR was performed using the 7900 HT Fast Real Time PCR system (SDS version 2.3 software) or CFX384 Touch Real-Time PCR Detection System (Bio-Rad, Segrate, Italy) using commercially available primers (TaqMan Gene Expression Assays; Thermo Fisher Scientific, Monza, Italy) listed in Appendix A. Relative gene expression was calculated as described [46].

### 4.7. Western Blot Analysis

Protein concentration was measured using the Bradford assay (Bio-Rad). The samples were fractionated on SDS-PAGE and transferred to nitrocellulose membranes (Bio-Rad) [46]. Full details are in Appendix A.

### 4.8. Confocal Microscopy

Cells were cultured on glass cover slips in 24-well plates for 24 h and then stained and images analyzed as described in Appendix A.

### 4.9. FACS Analysis

Data were acquired using a BD FACSCelesta Flow Cytometer and data processed with DIVA version 10 (BD Biosciences, San Jose, CA, USA) and FlowJo version 10.01 softwares (TreeStar, Ashland, OR, USA).

### 4.10. Seahorse Metabolic Experiments

Real-time measurements of oxygen consumption rate (OCR) and extracellular acidification rate (ECAR) were made using an XFe96 Extracellular Flux Analyzer (Agilent Technologies, Santa Clara, CA, USA), using Mito Stress Test and Glycolysis Stress Test as described in Appendix A.

### 4.11. Lactate Dehydrogenase (LDH), Glyceraldehyde 3-Phosphate Dehydrogenase (GAPDH) and Glucose 6-Phosphate Dehydrogenase (G6PD) Activity

Cells were re-suspended at 1 × 10^5^ cells/mL in 0.2 mL of 82.3 mM triethanolamine phosphate hydrochloride (TRAP, pH 7.6) for LDH and GAPDH assays, in 0.1 M Tris/0.5 mM EDTA pH 8.0 for G6PD assay, and sonicated on ice with two 10 s bursts. The enzymatic activities were measured spectrophotometrically by monitoring the oxidation of NADH (LDH, GAPDH activity) or the reduction of NADP^+^ (G6PD activity), as described previously [47].

### 4.12. Electron Transport Chain (ETC) Flux and Mitochondrial ATP

The electron flux from complex I to complex III, taken as index of the mitochondrial respiratory activity, was measured spectrophotometrically on isolated mitochondria as detailed in [48]. The amount of ATP in mitochondrial extracts was measured with the ATP Bioluminescent Assay Kit (Sigma-Merck, Milan, Italy).

### 4.13. Pentose Phosphate Pathway (PPP) Flux

The activity of PPP was measured by radiolabeling cells with 2 µCi [1-14C] glucose or [6-14C] glucose (Dupont-New England Nuclear, Boston, MA, USA). The metabolic fluxes through the PPP + tricarboxylic acid cycle and the tricarboxylic acid cycle were measured by detecting the amount of 14CO2 developed from [6-14C] glucose or [1-14C] glucose in 1 h, respectively [47].

### 4.14. Colony Formation Assay

Cells (500/well) were seeded into 6-well plates and cultured for 10–12 days in a complete medium with or without BRAFi (0.5 µM). Cells were then fixed with 4% PFA (10 min, RT) and stained with crystal violet (20 min, RT, in the dark). Images were acquired using Axio Observer Z1 microscope (Zeiss, Milan, Italy), using a 20×/0.4NA objective. The percentage of occupied colony areas was calculated with ImageJ/Fiji software.

### 4.15. Soft Agar Colony Formation Assay

The Soft Agar Colony Formation Assay was performed as described previously [49]. Briefly, the Soft Agar Colony Formation Assay was performed creating two layers of agar. A single-cell suspension in medium of A375 and M14 GFP and NAMPT/GFP variants was plated into 6-well plates (5000–7500 cells/well) containing 0.3% low melting agarose and solidified 0.6% agarose. Cells in agar were incubated at 37 °C in a humidified environment for ~20 days. 100 μL of medium were added twice weekly to each well to prevent desiccation. Colonies were stained by adding 200 μL of nitroblue tetrazolium chloride solution (Sigma)/well and incubating plates overnight at 37 °C. Images were acquired using Axio Observer Z1 microscope (Zeiss) equipped with a 20×/0.4NA objective. Image analysis was performed using Z-stacks projection and the size of colonies calculated using ImageJ/Fiji software.

### 4.16. Signaling Experiments

A375 and M14 variants were starved for 24 h in medium without serum, then 10% FCS was added for the indicated time points at 37 °C. Cells were then lysed and ERK1/2 phosphorylation status was analyzed by western blot.

### 4.17. Invasion Assay

Invasion assays were performed using 8 μM pore Boyden chambers pre-coated with Matrigel 0.5 mg/mL (all from Corning, Corning, NY, USA). After 24 h, cells penetrated in the filter were stained with crystal violet and examined using bright-field microscopy and analyzed with ImageJ/Fiji software, as previously described [22].

### 4.18. Zymography Assay

A375 and M14 variants were starved in medium without serum for 24 h. Cells were lysed as previously described [50] and resolved using Novex Zymogram Plus gels (Thermo Fisher Scientific), following the indicated protocol of gel renaturing and developing. Lastly, the gels were stained with Blu Comassie solution (Sigma). Areas of protease activity appear as clear bands against a dark background and were quantified using ImageJ software.

### 4.19. Side population Analysis

Side population (SP) analysis was performed as described previously [51]. Briefly, 5 × 105 cells/tube were incubated in 500 μL of RPMI-1640 with 10% of FCS (1 h, 37 °C, protected from light) with Vybrant ™ DyeCycle ™ Violet Stain (DCV; 5 μM; Thermo Fisher Scientific, Monza, Italy). Samples were analyzed by FACS, gating on cells that lost DCV corresponding to SP.

### 4.20. Xenograft Models

All 6–8-week old male NOD/SCID/gamma chain-/-(NSG) mice were purchased by Charles River Laboratories International, Wilmington, MA, USA), and were maintained in a specific-pathogen-free (SPF) facility of the Molecular Biotechnology Center (MBC, UniTo, Italy). All related protocols were performed in compliance with the Guide for the Care and Use of Laboratory Animals. MM cells (5 × 10^6^) were resuspended in Matrigel^®^ (Corning) and injected into the right and left flank of 6–8-week old male NSG mice. Doxycycline (DOX, 0.1 mg/mL, Sigma), to induce or to silence NAMPT, was added biweekly in drinking water, as reported in the schemes in Appendix A. The tumor size was measured weekly using calipers in two dimensions to generate a tumor volume using the following formula: 0.5 × (length × width^2^). After 4–5 weeks mice were euthanized and tumor mass weight and volume were measured.

This research has been approved by the Italian Ministry of Health (authorization code 1179/2016-PR, approved 15 December 2016).

### 4.21. TCGA Analysis

Genomic data shown in this paper are in whole or part based upon data generated by The Cancer Genome Atlas (TCGA) Research Network. Clinical and genomic data of TCGA cancer samples and patients (release 2016_02_28) were downloaded from the Broad TCGA GDAC site, by means of firehose get Version: 0.4.1.

### 4.22. Statistical Analysis

Statistical comparisons were performed using Graph Pad Prism version 7.0 (Graph Pad Software Inc., La Jolla, CA, USA). Statistical significance was determined by two-tailed Mann–Whitney U or unpaired/paired Student’s *t* test and one-way ANOVA test. Full details of statistical TCGA analyses are included in SM.

Unless otherwise indicated, data in the Figures are presented as the mean ± SEM. For all statistical tests, the 0.05 level of confidence was accepted for statistical significance. Significance was represented as: * *p* ≤ 0.05, ** *p* ≤ 0.01, *** *p* ≤ 0.001 and **** *p* ≤ 0.0001.

## 5. Conclusions

The identification and possible targeting of the main factors that regulate the onset of BRAFi-resistance mechanisms in MM represents an urgent clinical need for these subset of patients. Herein, we provided complete and definitive evidence that NAMPT may be considered as a main driving force for drug resistance and melanoma progression and aggressiveness. These findings open the way to future clinical experimentations combining NAMPT inhibitors and BRAF-targeted therapies. 

## Figures and Tables

**Figure 1 cancers-12-03855-f001:**
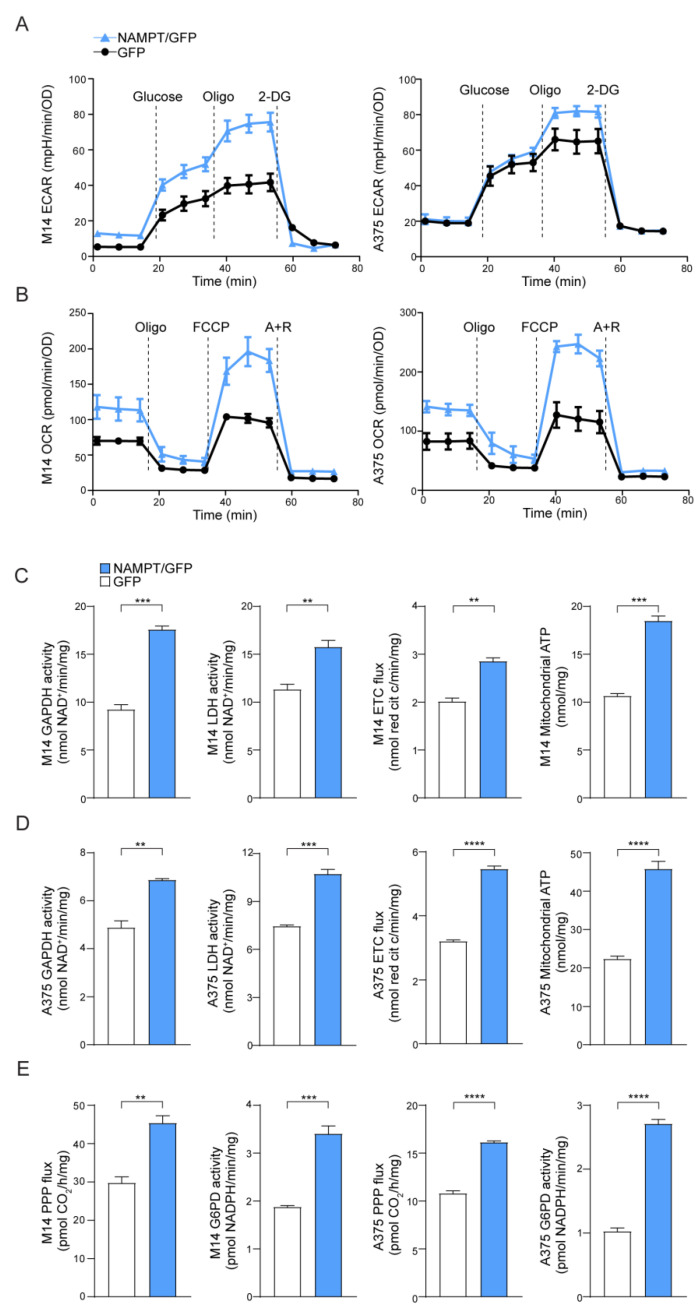
Nicotinamide phosphoribosyltransferase (NAMPT) over-expression boosts metabolic pathways. (**A**) Extracellular acidification rate (ECAR) and (**B**) O2 consumption rate (OCR) measured in M14 and A375 NAMPT/GFP (light blue) vs. GFP (black) variants using the Seahorse XFe Analyzer (representative profile of 3 independent experiments). Oligo: oligomycin, 2-DG: 2-Deoxy-D-glucose, FCCP: carbonyl cyanide p-trifluoromethoxyphenylhydrazone, A + R: antimycin + rotenone. (**C**,**D**) Histograms showing metabolic measurements of GAPDH and LDH activity (nmol NAD + /min/mg protein), electron transport chain (ETC) flux (nmol red cit c/min/mg protein), and mithocondrial ATP (nmlo/mg protein) in M14 (**C**) and A375 (**D**), comparing NAMPT/GFP vs. GFP variants. Cumulative data of 3 independent experiments. Unpaired *t* test. (**E**) Pentose phosphate pathway flux (PPP, pmolCO2/h/mg protein) and G6PD activity (nmol/min/mg protein) in M14 and A375 comparing NAMPT/GFP vs. GFP variants. *N* = 3, unpaired t test. Data in the Figure are presented as the mean ± SEM. Significance was represented as: ** *p* ≤ 0.01, *** *p* ≤ 0.001 and **** *p* ≤ 0.0001.

**Figure 2 cancers-12-03855-f002:**
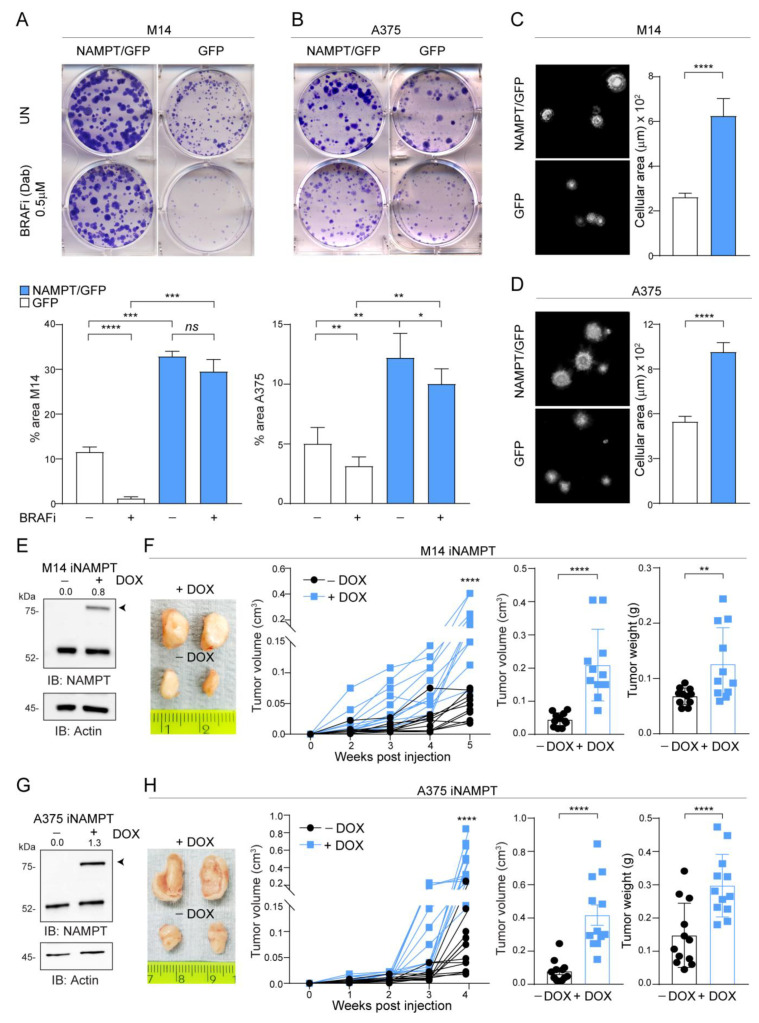
NAMPT over-expression forces MM cells growth capacity in vitro and in vivo. (**A**,**B**) Colony-forming ability of M14 (**A**) and A375 (**B**) NAMPT/GFP in comparison with GFP control cells in untreated (UN) or treated with the indicated dose of dabrafenib (BRAFi) every 72 h for 12 days. Cells were stained with crystal violet and representative images are shown. Below the images, histograms show the cumulative quantification of the percentage (%) area with colonies at the end of the 12-days period (at least 3 independent experiments performed in triplicates, Mann–Whitney and paired t test). (**C**,**D**) Representative images of soft-agar colony-forming ability of M14 (**C**) and A375 (**D**) NAMPT/GFP in comparison with GFP control cells in complete medium for 20–25 days. On the right, cumulative data of the area with colonies are shown (measured from at least 3 different fields of 3 pictures of 3 independent experiments, Mann–Whitney test). (**E**,**G**) Western blots showing inducible NAMPT/GFP expression (the band indicated with the arrow) upon doxycycline (DOX, 1 µg/mL, 72 h) treatment of NAMPT-inducible (i) M14 (**E**) or iA375 (**G**) cell lines. Actin was used as loading control. Densitometry intensity ratios of NAMPT/GFP bands (indicated with arrows) over actin were reported. (**F**,**H**) Representative tumor masses derived from NSG mice xenografted with iM14 (**F**) and iA375 (**H**) variants (5 × 10^6^ cells, injected subcutaneously) and treated or not with DOX (0.1 mg/mL, twice a week orally) for 4–5 weeks. At least 8–10 animals for each condition were evaluated. Graphs represent tumor volume growth kinetics over a period of 4–5 weeks post-injection, cumulative tumor volume and tumor weight at sacrifice for both cell lines. Mann–Whitney test. Data in the Figure are presented as the mean ± SEM. Significance was represented as: * *p* ≤ 0.05, ** *p* ≤ 0.01, *** *p* ≤ 0.001 and **** *p* ≤ 0.0001. Uncropped Western Blot Images in Appendix A.

**Figure 3 cancers-12-03855-f003:**
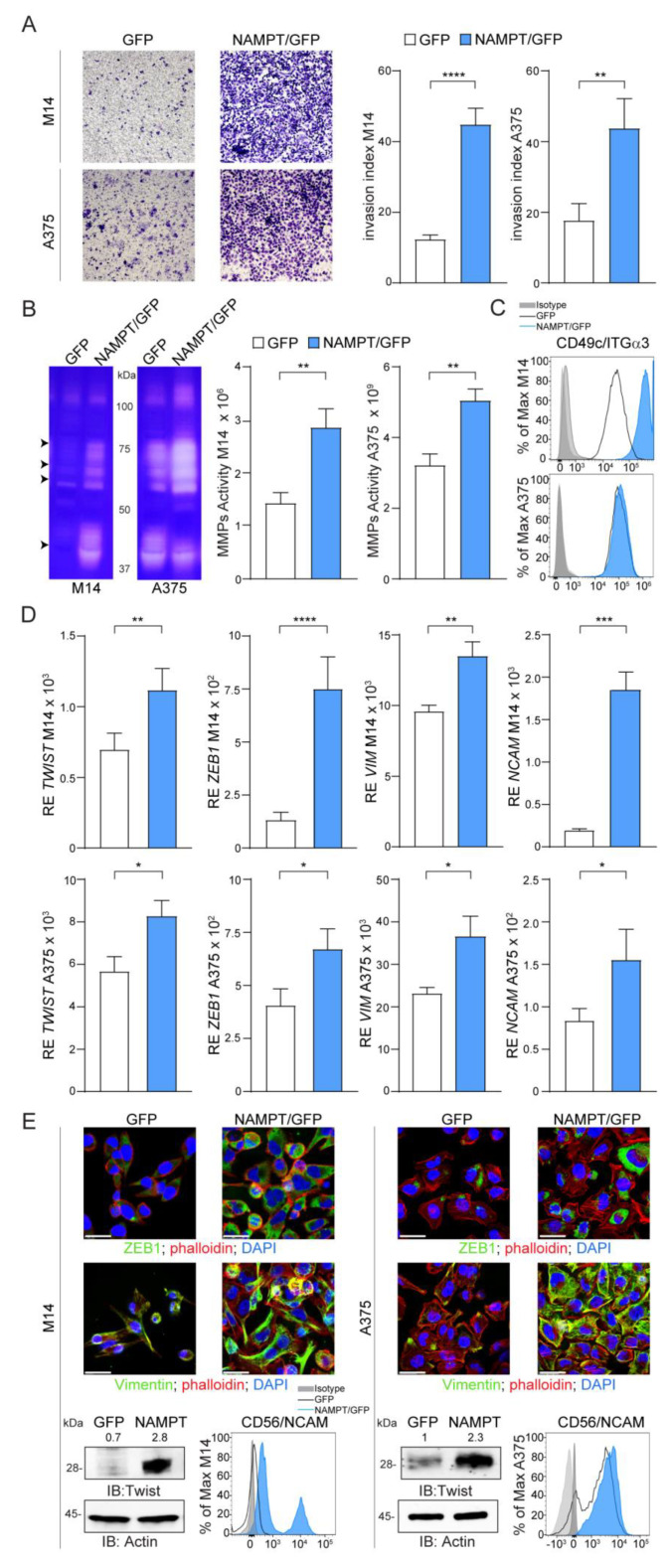
NAMPT over-expression induces a phenotype switch towards invasive and mesenchymal features. (**A**) Representative images (×10 magnification) of invasion assay in Matrigel for 24 h of M14 and A375 cell lines comparing NAMPT/GFP and GFP variants. Histograms on the right represent cumulative data of invasion assays (at least 6 independent experiments). Invasion index was calculated as the number of cells penetrated in the presence of the chemoattractant/number of cells penetrated without chemoattractant. (**B**) Gel zymography analysis of cell extracts derived from M14 and A375 NAMPT/GFP and GFP variants. Arrows indicate the most up-regulated metalloproteinase (MMP) activities at molecular weights compatible with MMP-9 (92 kDa), MMP-2 (72 kDa), MMP-1 and MMP-3 (55-45 kDa). Histograms on the right show cumulative data of MMPs activity obtained from gel zymography analysis in both cell lines variants (at least 4 independent experiments, Mann–Whitney test). (**C**) Differential expression of integrin-α3 (ITGα3) in M14 and A375 NAMPT/GFP and GFP variants detected by FACS analysis. (**D**,**E**) qRT-PCR analysis (**D**) or representative confocal (green fluorescence, original magnification 63×, scale bar: 25 µm), western blot and FACS analysis, (**E**) showing expression of mesenchymal molecules including TWIST, ZEB1, VIM and NCAM. In qRT-PCR and western blot experiments, expression levels of the analyzed gene/protein were normalized over Actin. In the representative western blot densitometry intensity ratios of Twist bands over actin were reported. Results were obtained from at least 6 independent experiments. Unpaired *t* test. Data in the Figure are presented as the mean ± SEM. Significance was represented as: * *p* ≤ 0.05, ** *p* ≤ 0.01, *** *p* ≤ 0.001 and **** *p* ≤ 0.0001. Uncropped Western Blot Images in Appendix A.

**Figure 4 cancers-12-03855-f004:**
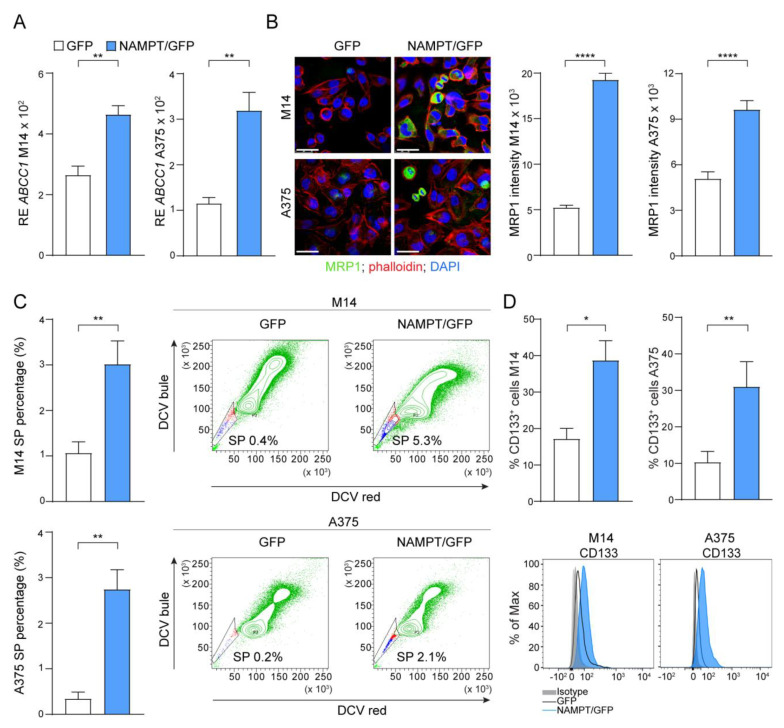
NAMPT over-expression forces stemness properties. (**A**) Histograms reporting cumulative mRNA expression levels of the ATP Binding Cassette Subfamily C Member 1 (*ABCC1*) in M14 and A375 cell lines comparing NAMPT/GFP and GFP variants. At least 5 independent RT-PCR experiments (Mann–Whitney test). (**B**) Confocal staining for ABCC1/MRP1 (green fluorescence) in M14 and A375 cell lines comparing NAMPT/GFP and GFP variants (original magnification 63×, scale bar: 25 µm). On the right, cumulative data of MRP1 fluorescence intensity measured from at least 3 different fields of 3 pictures of 3 independent confocal experiments (Mann–Whitney test). (**C**) Histograms showing cumulative results of at least 5 independent experiment to measure the percentage (%) of Side Population (SP) in M14 and A375 variants (Mann–Whitney test). Representative density plots show the increased % SP (the small staminal cell population, highlighted in the gates, that lost dye cycle violet DCV used in FACS staining) in NAMPT/GFP cells compared with GFP. (**D**) Differential expression of the % of CD133^+^ cells in M14 and A375 NAMPT/GFP and GFP variants detected by FACS analysis (6 independent experiments, Mann–Whitney test). Representative histogram plots are reported below the graphs. Data in the Figure are presented as the mean ± SEM. Significance was represented as: * *p* ≤ 0.05, ** *p* ≤ 0.01, and **** *p* ≤ 0.0001.

**Figure 5 cancers-12-03855-f005:**
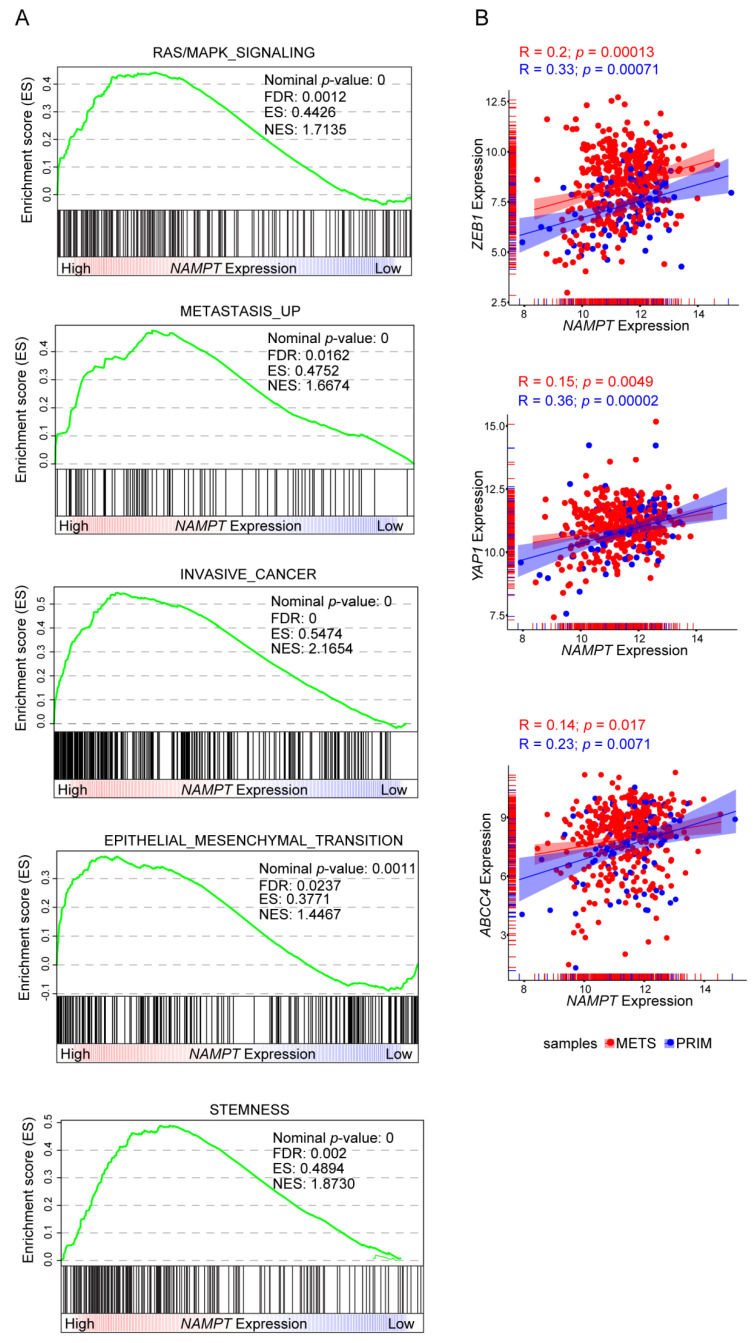
Positive correlations between *NAMPT* expression and cancer aggressive features in gene set enrichment analysis of TCGA melanoma patients cohort. (**A**) GSEA plots of differentially expressed genes, belonging to the indicated categories, between high and low *NAMPT* expressing Skin Cutaneous Melanoma (SKCM) samples of the TCGA cohort. All the represented gene sets positively correlates with NAMPT expression at a false discovery rate (FDR < 0.05). Enrichment score (ES), normalized enrichment score (NES). (**B**) Scatter plot correlating *NAMPT* expression and *ZEB1*, *YAP1*, and *ABCC4* respectively. Each dot represents a sample of the TCGA SKCM cohort, colored in red if metastasis (METS) derived and in blue if primary (PRIM) tumor-derived. Pearson correlation (R) and *p*-value are shown. The expression of all three genes positively correlates with *NAMPT* expression.

**Figure 6 cancers-12-03855-f006:**
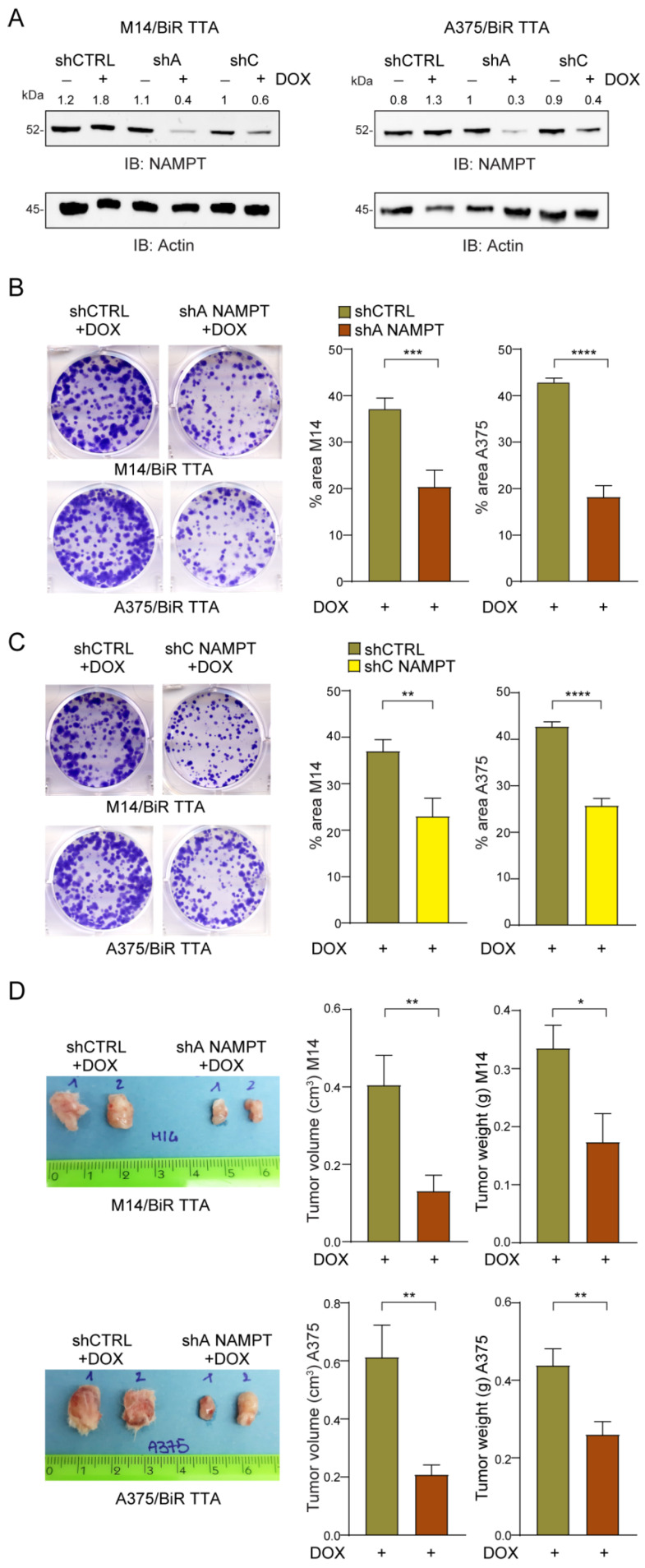
*NAMPT* silencing reduces tumor growth of/BiR cells in vitro and in vivo. (**A**) Representative NAMPT western blot (**B**) in M14/BiR TTA and A375/BiR TTA cell lines with a DOX-dependent inducible NAMPT silencing (using two different shRNAs NAMPT sequences **A**,**C**) in comparison with shCTRL cells. Densitometry intensity ratios of NAMPT bands over actin were reported. (**B**,**C**) Colony-forming ability of shA NAMPT (**B**) or shC NAMPT (**C**) in comparison with shCTRL M14/BiR TTA and A375/BiR TTA cell lines treated with DOX (1 µg/mL) every 72 h for 12 days. Cells were stained with crystal violet and representative images are shown. On the right, histograms show the cumulative quantification of the percentage (%) of the area with colonies at the end of the 12-days period (at least 5 independent experiments, Mann–Whitney test). (**D**) Representative tumor masses derived from NSG mice xenografted with M14/BiR TTA and A375/BiR TTA cell lines inducible shA NAMPT and shCTRL variants (5 × 10^6^ cells, injected subcutaneously) and treated with DOX (0.1 mg/mL, twice a week orally) 24 h after injection for 4–5 weeks. At least 8–10 animals under each condition were evaluated. Graphs represent cumulative tumor volume and tumor weight measurements when mice were sacrificed, in both cell lines (Mann–Whitney test). Data in the Figure are presented as the mean ± SEM. Significance was represented as: * *p* ≤ 0.05, ** *p* ≤ 0.01, *** *p* ≤ 0.001 and **** *p* ≤ 0.0001. Uncropped Western Blot Images in Appendix A.

**Figure 7 cancers-12-03855-f007:**
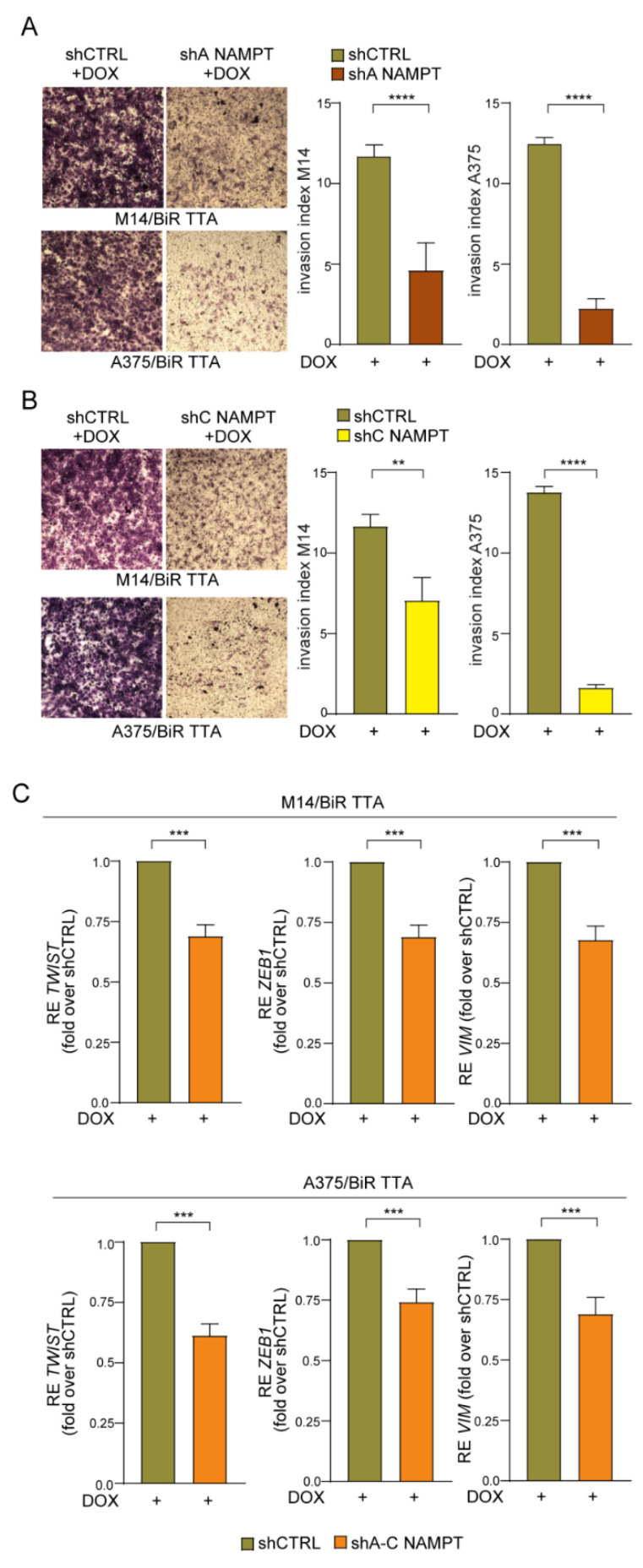
*NAMPT* silencing reverts the aggressiveness of/BiR cells. (**A**,**B**) Representative images (×10 magnification) of invasion assay in matrigel of M14/BiR TTA and A375/BiR TTA cell lines inducible shA NAMPT (**A**) and shC NAMPT (**B**) in comparison with shCTRL cells treated with DOX (1 µg/mL) for 24 h before performing assay. On the right histograms represent cumulative data of invasion assays (at least 5 independent experiments). Invasion index was calculated as: number of cells penetrated in the presence of the chemoattractant/number of cells penetrated without chemoattractant. Mann-Whitney test. (**C**) qRT-PCR analysis showing expression of mesenchymal markers including *TWIST*, *ZEB1* and *VIM* in M14/BiR TTA and A375/BiR TTA cell lines inducible shNAMPT vs. shCTRL cells treated with DOX (1 µg/mL) for 24 h. Results show the reduction of expression of the genes in NAMPT silenced cells (sh **A**–**C**) compared with shCTRL (results are represented as fold-over shCTRL of at least 8 independent experiments, Wilcoxon test). Data in the Figure are presented as the mean ± SEM. Significance was represented as: ** *p* ≤ 0.01, *** *p* ≤ 0.001 and **** *p* ≤ 0.0001.

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
