# Peer review of "NAMPT Over-Expression Recapitulates the BRAF Inhibitor Resistant Phenotype Plasticity in Melanoma"

_cancers, 2020, doi:10.3390/cancers12123855_

Round 1
Reviewer 1 Report
I have carefully evaluated the experiments and relative answers that the authors have highlighted to complite my review and I find them satisfactory. I warmly recommend the publication of manuscript by Audrito et al. entitled "NAMPT Over-Expression Recapitulates the BRAF 3 Inhibitor Resistant Phenotype Plasticity in Melanoma" on Cancers
Reviewer 2 Report
The authors have adequately addressed all of my concerns.
This manuscript is a resubmission of an earlier submission. The following is a list of the peer review reports and author responses from that submission.
Round 1
Reviewer 1 Report
In this manuscript, the authors expand on their prior findings that BRAFi-resistant melanomas overexpress NAMPT, and they describe the role of NAMPT in the development of BRAFi-resistance in BRAF-mutated melanoma. Using two models, they show that overexpression of NAMPT in BRAF-mutated melanoma cell lines changes a number of features of these cells, namely their metabolic activity, ERK1/2 phosphorylation, invasiveness/aggressiveness, mesenchymal properties and stemness. Using GSEA data, they demonstrate that melanomas with higher NAMPT expression are more likely to have invasive, metastatic, mesenchymal and stem-like gene signatures. Finally in two models of BRAFi-resistant melanoma, using inducible shRNA against NAMPT, they show that NAMPT silencing reduces colony formation and tumor growth in vivo, as well as invasiveness and mesenchymal phenotype.
The conclusions drawn from each set of experiments are generally well supported and the experiments themselves are well controlled. A main concern is that the central assertion is that NAMPT causes BRAFi resistance in melanoma, but the data are more suggestive of an association versus causation. There are several experiments that would better support this claim that should be performed. In addition, text editing could make the conclusions more in line with the actual data presented.
Major comments:
- The experiments shown in Figure 2 A and 2B are the only ones that explicitly are designed to show that overexpression of NAMPT induces drug resistance to BRAFi (dabrafenib) in these cell lines, a central premise to this manuscript. While this appears to be true for the M14 cell line, the images for A375 do not clearly show a difference in colony formation between the control and overexpression cell lines when treated with dabrafenib. Although the quantification suggests there is a difference between the 2 cell conditions, visually the two conditions look incredibly similar. The data would be more compelling if the differences were clearer and if additional cell lines and/or BRAFi were tested to support this central claim.
- The manuscript does not establish whether the changes described in Figures 1, 2, 3, and 4 are due to increased NMN or another consequence of higher NAMPT expression. It would provide greater mechanistic insight to know whether the addition of NMN alone recapitulates the findings in these figures or if it something else about the NAMPT overexpression that contributes.
- The experiments presented in Figure 6 (shRNA against NAMPT) show that in the BRAFi-resistant melanoma lines, depletion of NAMPT decreases colony formation and tumor growth and the authors assert that this is evidence that NAMPT is responsible for BRAFi resistance. However, depletion of NAMPT has been shown to reduce cellular viability in many models of cancer and without data from a control melanoma cell line that is not BRAFi resistant and is not similarly affected by NAMPT depletion, this claim is not adequately supported.
- Given that a final conclusion of the manuscript is that NAMPTi could be used in combination with BRAFi to mitigate the development of resistance, experiments testing whether NAMPTi recapitulate the phenotypes of the shRNA against NAMPT should be included.
Minor comments:
- The authors describe the findings in Figure 1 as cells showing increased metabolic "fitness" and "performance." Certainly there is evidence of increased metabolic activity but the use of these other terms suggest some sort of advantageous change, which there is not evidence to support.
- Figure S1A shows only RNA expression for GLUT1. Showing protein expression as well would strengthen the functional implications of this finding.
- The language on page 9, lines 218-219, implies causation and specificity to BRAFi-resistance that overstates what the data show.
- Figure 5 lacks data on RAS/MAPK gene signatures, but the text suggests it is shown (page 10, line 222). The text should be edited or the relevant data should be added.
- The Figure 5 legend refers to "SKCM" - what does this mean? This should be clarified.
- The concluding sentence in the results section (page 15, line 313) refers to the melanocyte transitioning to a cancer cell, but there is no evidence for role of NAMPT in tumor initiation presented in this manuscript. This should be edited.
- The discussion makes several claims that are not completely supported by the data presented including that (1) the impact of increased phosphorylation of ERK1/2 acts as a positive feedback to sustain oncogenic signaling and that (2) combining NAMPTi and BRAFi is a good translational strategy, which is not tested (as mentioned above).
- There are a number of grammatical/typographical errors throughout the manuscript that should be corrected with the next re
Author Response
Reviewer 1
In this manuscript, the authors expand on their prior findings that BRAFi-resistant melanomas overexpress NAMPT, and they describe the role of NAMPT in the development of BRAFi-resistance in BRAF-mutated melanoma. Using two models, they show that overexpression of NAMPT in BRAF-mutated melanoma cell lines changes a number of features of these cells, namely their metabolic activity, ERK1/2 phosphorylation, invasiveness/aggressiveness, mesenchymal properties and stemness. Using GSEA data, they demonstrate that melanomas with higher NAMPT expression are more likely to have invasive, metastatic, mesenchymal and stem-like gene signatures. Finally in two models of BRAFi-resistant melanoma, using inducible shRNA against NAMPT, they show that NAMPT silencing reduces colony formation and tumor growth in vivo, as well as invasiveness and mesenchymal phenotype.
The conclusions drawn from each set of experiments are generally well supported and the experiments themselves are well controlled. A main concern is that the central assertion is that NAMPT causes BRAFi resistance in melanoma, but the data are more suggestive of an association versus causation. There are several experiments that would better support this claim that should be performed. In addition, text editing could make the conclusions more in line with the actual data presented.
Reply: Thanks for the positive evaluation of our paper
Major comments:
- The experiments shown in Figure 2 A and 2B are the only ones that explicitly are designed to show that overexpression of NAMPT induces drug resistance to BRAFi (dabrafenib) in these cell lines, a central premise to this manuscript. While this appears to be true for the M14 cell line, the images for A375 do not clearly show a difference in colony formation between the control and overexpression cell lines when treated with dabrafenib. Although the quantification suggests there is a difference between the 2 cell conditions, visually the two conditions look incredibly similar. The data would be more compelling if the differences were clearer and if additional cell lines and/or BRAFi were tested to support this central claim.
Reply: We thank the reviewer for this comment. We previously demonstrated that NAMPT over-expressing cells M14 and A375 growth faster and acquire BRAFi-resistance earlier compared to GFP-control infected cells. These data are included in a paper published in 2018 [Audrito et al. JNCI 2018]. Here we focused on understanding the role of NAMPT in promoting the acquisition of a resistant and aggressive phenotype, analyzing different functional readouts including the capacity of these cells to form colonies in 2D and 3D.
The colony forming assay was performed at least 3 times (independent experiments performed in triplicate) for the original submission. The plates were automatically analyzed using ImageJ to quantify the area occupied by cells. We are enclosing the excel file with raw data containing at least 9 measures per condition. Statistical analysis using Mann Whitney test (to compare NAMPT vs GFP cells) or paired t test (to compared treated vs untreated condition) indicates statistical significant differences for both M14 and A375 cell lines. Following the reviewer’s recommendations, we changed the image of colony-forming assay of A375. In the image now included the difference between BRAFi-treated NAMPT-overexpressing cells compared to GFP is more visible. Moreover, following the reviewer’s suggestion, we decided to repeat the colony-formation assay using a different BRAFi (Vemurafenib). This experiment was performed three times (independent experiments in triplicate wells) over a 7-day course, showing a similar difference in the ability to form colonies, confirming an intrinsic resistance to BRAFi in NAMPT-overexpressing cells. These data are now included in the revised version of the text and in revised Supplementary Figure 2.
- The manuscript does not establish whether the changes described in Figures 1, 2, 3, and 4 are due to increased NMN or another consequence of higher NAMPT expression. It would provide greater mechanistic insight to know whether the addition of NMN alone recapitulates the findings in these figures or if it something else about the NAMPT overexpression that contributes.
Reply: We thank the reviewer for raising this is very interesting point, aimed at asking the question of whether the observed effects are due to NAMPT enzymatic activity, which leads to increased NAD availability. To address this question and following the Reviewer’s suggestion we added NMN, the water-soluble product of the NAMPT-controlled reaction, and studied cellular phenotype. To this aim, we selected experiments with a relatively short read-out, considering that we do not know stability and metabolism of NMN, as well as bioavailability inside the cell, when the compound is added in the extracellular medium. In any case, when looking at expression of genes related to a mesenchymal phenotype, NMN administration (added once at the beginning of 24-hour cultures) resulted in up-regulation of ZEB1 and its target VIM in both A375 and M14 (VIM up-regulation is significantly higher only in M14 cells, while there is a trend in A375, see revised Supplementary Figure 5). Consistently, cells cultured in the presence of NMN 1mM for 24 hours migrated slightly but significantly better than cells cultured without NMN. Based on these results, it is therefore possible to speculate that the aggressive phenotype of NAMPT/GFP A375 and M14 cells is due to increased enzymatic activity and ultimately to increased NAD levels, which in turn regulate NAD-consuming enzymes. These aspects are now better addressed in the revised Discussion.
It is however important to point out that NAMPT can also have extracellular effects, which are described to be enzyme-independent and likely due to signaling activation through TLR4 [Camp et al. Sci. Rep. 2015; Audrito et al Front. Oncol 2020], which is expressed by both cell lines. In this context, we documented presence of NAMPT in the supernatants of both cell lines, with increasing concentrations in BRAF-resistant cell lines (BiR) [Audrito et al. Oncotarget 2018] and also in the supernatants of NAMPT-overexpressing cells.
Moreover, the results included in a paper published at the end of 2017 by Lucena-Cacace et al, analyzing the effects of NAMPT over-expression in colon cancer, showed that NMN treatment only partially recapitulate the effect of NAMPT-overexpression in colon cancer model [Lucena-Cacace et al, Clin Cancer Res 2017]. These results are in line with our evidence in the melanoma model.
We included in the discussion section these considerations.
- The experiments presented in Figure 6 (shRNA against NAMPT) show that in the BRAFi-resistant melanoma lines, depletion of NAMPT decreases colony formation and tumor growth and the authors assert that this is evidence that NAMPT is responsible for BRAFi resistance. However, depletion of NAMPT has been shown to reduce cellular viability in many models of cancer and without data from a control melanoma cell line that is not BRAFi resistant and is not similarly affected by NAMPT depletion, this claim is not adequately supported.
Reply: We agree with the reviewer that depletion of NAMPT, or perhaps depletion of NAD as a consequence of NAMPT inhibition reduces viability in several cell models, but it certainly isn’t a universal finding. In fact, response to NAMPTi is dictated by the expression of other NAD-synthesizing enzymes, including NAPRT and NRK, which synthesize NAD from dietary sources [Chowdhry et al, Nature 2019]. Our observation, obtained by comparing A375 and M14 in their original conformation to the BiR resistant version, indicates that BiR cells become uniquely dependent on NAMPT activity as they express low to undetectable levels of other NAD-biosynthetic enzymes. This was also shown in vivo using NAMPT inhibitors in xenograft models [Audrito et al. JNCI 2018]. Now, we show that genetic ablation of this gene renders BiR cells not viable, while inducible silencing of NAMPT reverts the aggressive phenotype of BiR cells. This is what this paper focuses on.
- Given that a final conclusion of the manuscript is that NAMPTi could be used in combination with BRAFi to mitigate the development of resistance, experiments testing whether NAMPTi recapitulate the phenotypes of the shRNA against NAMPT should be included.
Reply: Thanks for this observation. We discussed this point suggesting this as possible novel therapeutic strategy for MM patients because we know that resistant cells become uniquely dependent on NAMPT activity as we showed in vitro and in xenograft models using NAMPTi [Audrito et al. JNCI 2018], and now we know that NAMPT is a driver molecule to promote MM aggressive phenotype. Our hypothesis is that a combination therapy of NAMPTi and BRAFi could be avoid the onset of resistance. Obviously this idea should be tested in a pilot clinical trial.
Minor comments:
- The authors describe the findings in Figure 1 as cells showing increased metabolic "fitness" and "performance." Certainly there is evidence of increased metabolic activity but the use of these other terms suggest some sort of advantageous change, which there is not evidence to support.
Reply: Thank you for these suggestions. We changed “fitness” with metabolic fluxes
- Figure S1A shows only RNA expression for GLUT1. Showing protein expression as well would strengthen the functional implications of this finding.
Reply: We totally agree with the reviewer. We added confocal microscopy to show GLUT1 protein levels in M14 and A375 NAMPT vs GFP variants. These data are now included in the revised version of the text and in revised Supplementary Figure 1.
- The language on page 9, lines 218-219, implies causation and specificity to BRAFi-resistance that overstates what the data show.
Reply: We agree with the reviewer’s consideration and we decide to delete this sentence.
- Figure 5 lacks data on RAS/MAPK gene signatures, but the text suggests it is shown (page 10, line 222). The text should be edited or the relevant data should be added.
Reply: Thanks a lot for noticing this mistake, which is amended in the revised Figure 5A.
- The Figure 5 legend refers to "SKCM" - what does this mean? This should be clarified.
Reply: Thank you, we added the complete name Skin Cutaneous Melanoma (SKCM) that is the cohort of melanoma of TCGA
- The concluding sentence in the results section (page 15, line 313) refers to the melanocyte transitioning to a cancer cell, but there is no evidence for role of NAMPT in tumor initiation presented in this manuscript. This should be edited.
Reply: Thank you for pointing this sentence. We agree with you and we re-phased the sentence in the revised version.
- The discussion makes several claims that are not completely supported by the data presented including that (1) the impact of increased phosphorylation of ERK1/2 acts as a positive feedback to sustain oncogenic signaling and that (2) combining NAMPTi and BRAFi is a good translational strategy, which is not tested (as mentioned above).
Reply: Thanks for underlining these two points of the discussion. Following your indications we made some changes in the revised text.
- There are a number of grammatical/typographical errors throughout the manuscript that should be corrected with the next re
Reply: Thank you. We have checked the manuscript text and corrected some errors.
Reviewer 2 Report
The manuscript by Audrito et al. entitled "NAMPT Over-Expression Recapitulates the BRAF 3 Inhibitor Resistant Phenotype Plasticity in Melanoma" reports a comprehensive and accurate investigation that shows how the overexpression of the NAMPT enzyme is correlated with an increase in the invasiveness and stemness of melanoma cancers cells.
In my opinion this work highlights an interesting point of view on the mechanisms underlying the greater invasiveness of tumors, but the reported results call for further information to fully understand the role of NAMPT. Indeed, it would be appropriate to demonstrate whether the effects of the overexpression of the enzyme, well described by the authors, are to be ascribed to its enzymatic activity or to the mere presence of the enzyme.
In particular:
- Do the increase of concentration of the enzymatic product NAD + or the decrease in NMN substrate correlate with the presented results?
- Do the authors also have evidence regarding the possibility that NAMPT is involved in the formation of protein-protein complexes with other involved proteins?
- As the authors propose the oncogenic function of NAMPT, they could comment on the possibility that this enzyme is translocated into the nucleus to perform that specific function, also in the light of recent published works that should then be cited (Grolla et al. JBC 2020).
At the end the Supplementary Figure 1 doesn't need the specification "A".
In brief, the paper is technically sound and I would recommend publication on Cancers with minor revision.
Author Response
Reviewer 2
The manuscript by Audrito et al. entitled "NAMPT Over-Expression Recapitulates the BRAF 3 Inhibitor Resistant Phenotype Plasticity in Melanoma" reports a comprehensive and accurate investigation that shows how the overexpression of the NAMPT enzyme is correlated with an increase in the invasiveness and stemness of melanoma cancers cells.
In my opinion this work highlights an interesting point of view on the mechanisms underlying the greater invasiveness of tumors, but the reported results call for further information to fully understand the role of NAMPT. Indeed, it would be appropriate to demonstrate whether the effects of the overexpression of the enzyme, well described by the authors, are to be ascribed to its enzymatic activity or to the mere presence of the enzyme.
Reply: We thank a lot the reviewer for the overall positive evaluation of our paper.
In particular:
- Do the increase of concentration of the enzymatic product NAD + or the decrease in NMN substrate correlate with the presented results?
Reply: We thank the reviewer for raising this is very interesting point, aimed at asking the question of whether the observed effects are due to NAMPT enzymatic activity, which leads to increased NAD availability. To address this question and following the Reviewer’s suggestion we added NMN, the water-soluble product of the NAMPT-controlled reaction, and studied cellular phenotype. To this aim, we selected experiments with a relatively short read-out, considering that we do not know stability and metabolism of NMN, as well as bioavailability inside the cell, when the compound is added in the extracellular medium. In any case, when looking at expression of genes related to a mesenchymal phenotype, NMN administration (added once at the beginning of 24-hour cultures) resulted in up-regulation of ZEB1 and its target VIM in both A375 and M14 (VIM up-regulation is significantly higher only in M14 cells, while there is a trend in A375, see revised Supplementary Figure 5). Consistently, cells cultured in the presence of NMN 1mM for 24 hours migrated slightly but significantly better than cells cultured without NMN. Based on these results, it is therefore possible to speculate that the aggressive phenotype of NAMPT/GFP A375 and M14 cells is due to increased enzymatic activity and ultimately to increased NAD levels, which in turn regulate NAD-consuming enzymes. These aspects are now better addressed in the revised Discussion.
It is however important to point out that NAMPT can also have extracellular effects, which are described to be enzyme-independent and likely due to signaling activation through TLR4 [Camp et al. Sci. Rep. 2015; Audrito et al Front. Oncol 2020], which is expressed by both cell lines. In this context, we documented presence of NAMPT in the supernatants of both cell lines, with increasing concentrations in BRAF-resistant cell lines (BiR) [Audrito et al. Oncotarget 2018] and also in the supernatants of NAMPT-overexpressing cells.
Moreover, the results included in a paper published at the end of 2017 by Lucena-Cacace et al, analyzing the effects of NAMPT over-expression in colon cancer, showed that NMN treatment only partially recapitulate the effect of NAMPT-overexpression in colon cancer model [Lucena-Cacace et al, Clin Cancer Res 2017]. These results are in line with our evidence in the melanoma model.
We included in the discussion section these considerations.
- Do the authors also have evidence regarding the possibility that NAMPT is involved in the formation of protein-protein complexes with other involved proteins?
- As the authors propose the oncogenic function of NAMPT, they could comment on the possibility that this enzyme is translocated into the nucleus to perform that specific function, also in the light of recent published works that should then be cited (Grolla et al. JBC 2020).
Reply: we thank the reviewer for these two interesting points to discuss. At this moment we do not have evidence of the interaction of NAMPT with other molecules in our human melanoma models, as suggested by Grolla et al in the indicated paper (quoted). However, we do not exclude that NAMPT could also translocate to the nucleus during BRAFi resistance development to increase NAD levels in that organelle and support the activity of sirtuins to modify gene expression or PARP to repair DNA damage. We added the reference of Grolla et al. JBC 2020 and discuss this hypothesis in the revised version of the paper.
At the end the Supplementary Figure 1 doesn't need the specification "A".
Reply: Thank you. In the revised Supplementary 1 we added the specification of the panel
In brief, the paper is technically sound and I would recommend publication on Cancers with minor revision.
Reply: Thank you very much.